

# High fructose exposure modifies the amount of adipocyte-secreted microRNAs into extracellular vesicles in supernatants and plasma

Adrián Hernández-Díazcouder[1,2], Javier González-Ramírez[3], Abraham Giacoman-Martínez[4], Guillermo Cardoso-Saldaña[5], Eduardo Martínez-Martínez[6], Horacio Osorio-Alonso[7], Ricardo Márquez-Velasco[2], José L. Sánchez-Gloria[2], Yaneli Juárez-Vicuña[2], Guillermo Gonzaga[7], Laura Gabriela Sánchez-Lozada[7], Julio César Almanza-Pérez[4] and Fausto Sánchez-Muñoz[2]

[1] Posgrado en Biología Experimental, Universidad Autónoma Metropolitana-Iztapalapa, Ciudad de México, México
[2] Departamento de Inmunología, Instituto Nacional de Cardiología Ignacio Chávez, Ciudad de México, México
[3] Laboratorio de Biología Celular, Facultad de Enfermería, Universidad Autónoma de Baja California Campus Mexicali, Mexicali, Baja California, Mexico
[4] Laboratorio de Farmacología, Departamento de Ciencias de la Salud, Universidad Autónoma Metropolitana-Iztapalapa, Ciudad de México, México
[5] Departamento de Endocrinología, Instituto Nacional de Cardiología Ignacio Chávez, Ciudad de México, México
[6] Laboratorio de Comunicación Celular y Vesículas Extracelulares, Instituto Nacional de Medicina Genómica, Ciudad de México, México
[7] Departamento de Fisiopatología Cardio-Renal, Instituto Nacional de Cardiología Ignacio Chávez, Ciudad de México, México

Corresponding author
Fausto Sánchez-Muñoz,
fausto.sanchez@cardiologia.org.mx,
fausto22@yahoo.com

## ABSTRACT

**Background**. High fructose exposure induces metabolic and endocrine responses in adipose tissue. Recent evidence suggests that microRNAs in extracellular vesicles are endocrine signals secreted by adipocytes. Fructose exposure on the secretion of microRNA by tissues and cells is poorly studied. Thus, the aim of this study was to evaluate the effect of fructose exposure on the secretion of selected microRNAs in extracellular vesicles from 3T3-L1 cells and plasma from Wistar rats.

**Methods**. 3T3-L1 cells were exposed to 550 µM of fructose or standard media for four days, microRNAs levels were determined in extracellular vesicles of supernatants and cells by RT-qPCR. Wistar rats were exposed to either 20% fructose drink or tap water for eight weeks, microRNAs levels were determined in extracellular vesicles of plasma and adipose tissue by RT-qPCR.

**Results**. This study showed that fructose exposure increased the total number of extracellular vesicles released by 3T3-L1 cells ($p = 0.0001$). The levels of miR-143-5p were increased in extracellular vesicles of 3T3-L1 cells exposed to fructose ($p = 0.0286$), whereas miR-223-3p levels were reduced ($p = 0.0286$). Moreover, in plasma-derived extracellular vesicles, miR-143-5p was higher in fructose-fed rats ($p = 0.001$), whereas miR-223-3p ($p = 0.022$), miR-342-3p ($p = 0.0011$), miR-140-5p ($p = 0.0129$) and miR-146b-5p ($p = 0.0245$) were lower.
**Conclusion**. Fructose exposure modifies the levels of microRNAs in extracellular vesicles in vitro and in vivo. In particular, fructose exposure increases miR-143-5p, while decreases miR-223-3p and miR-342-3p.

## INTRODUCTION

Over the past decades, sucrose and high fructose corn syrup have been used as the main fructose source in processed foods, including sweetened beverages. The high fructose intake promotes several metabolic abnormalities, including glucose intolerance, hypertension, hyperlipidemia, non-alcoholic fatty liver disease, chronic low-grade inflammation, adipose tissue expansion, and an imbalance in adipokine secretion from adipose tissue (*Legeza et al., 2017*; *Hernández-Díazcouder et al., 2019*). Moreover, it has been reported that high fructose exposure affects adipose tissue through mechanisms involving adipocyte function in humans, murine, and cellular models (*Zubiría et al., 2016*; *Pu & Veiga-Lopez, 2017*; *Hernández-Díazcouder et al., 2019*). Although the adipocytic cellular response to high fructose exposure has been explored, its endocrine activity has not been fully explored.

Several studies have demonstrated that adipose tissue produces adipokines that regulate lipid storage via endocrine and paracrine signals (*Perez-Diaz et al., 2018*; *Hörbelt et al., 2019*). For example, polymerase I and transcription release factor is secreted from adipocytes and it induces lipid accumulation in both hepatocytes and adipocytes (*Perez-Diaz et al., 2018*). Recently, microRNAs (miRNAs) have emerged as novel endocrine and paracrine signals (*Peng & Wang, 2018*). Adipocytes secrete miRNAs in exosomes and microvesicles, which are two subtypes of extracellular vesicles (EVs) that differ in size and molecular content (*Connolly et al., 2015*; *Durcin et al., 2017*). Moreover, it has been described that adipocytes release two types of EVs: small EVs (below 100 nm) and large EVs (100–200 nm) (*Durcin et al., 2017*). Recent studies have demonstrated that specific miRNAs transported in EVs participate as endocrine signals involved in the regulation of several cellular processes, including adipogenesis (*Huang-Doran, Zhang & Vidal-Puig, 2017*; *Thomou et al., 2017*). The transfer of miR-450a-5p through EVs activates adipogenesis in adipose tissue-derived stem cells (*Zhang et al., 2017*). Moreover, it has been demonstrated that fructose exposure modulates levels of miRNAs in EVs. For example, sucrose-fed rats showed an increase in levels of miRNAs related to inflammation in EVs such as miR-21-5p and miR-223-3p (*Brianza-Padilla et al., 2016*). However, there is no evidence that fructose regulates the levels of adipocyte-related miRNAs in EVs.

Several studies have shown that miRNAs regulate the translation of several genes in adipocytes (*Engin, 2017*; *Gebert & MacRae, 2019*). For example, it has been demonstrated that miR-140-5p and miR-450a-5p induce adipocyte differentiation (*Wang et al., 2016*; *Zhang et al., 2016*; *Zhang et al., 2017*). In addition to being involved in adipogenesis (*Jeong Kim et al., 2009*; *Wang et al., 2011*; *Wang et al., 2015b*; *Kim et al., 2012*; *Ahn et al.,*

*2013*; *Chen et al., 2014*; *Guan et al., 2015*; *Shi et al., 2015a*), miR-342-3p and miR-21-5p are related to the development of obesity. The miR-342-3p is up-regulated during the development of obesity (*Chartoumpekis et al., 2012*), and T2DM patients (*Collares et al., 2013*), and miR-21-5p is increased in adipose tissue of obese diabetic subjects (*Guglielmi et al., 2017*). Additionally, miR-223-3p is implicated in the inflammatory response (*Li et al., 2010*) and has been linked to metabolic alterations related to obesity (*Lu, Buchan & Cook, 2010*; *Zhuang et al., 2012*; *Sud et al., 2017*). miR-148a-5p is regulated by inflammatory cytokines and adipokines (*Shi et al., 2016*). Finally, miR-143-5p is related to the induction of insulin resistance by regulating insulin signaling (*Jordan et al., 2011*). Also, this miRNA promotes lipid accumulation in adipocytes (*Xie, Lim & Lodish, 2009*; *Wang et al., 2011*), and miR-146b-5p regulates glucose homeostasis by downregulation of IRS1 in preadipocytes (*Zhu et al., 2018*). A recent study found an increase in levels of miR-21-5p in the plasma of fructose-fed mice (*Engin, 2017*). In summary, this set of miRNAs display a clear role in obesity, T2DM, and adipogenesis (Table 1). Because these miRNAs are related to adipocyte functions, and it is known that fructose could modify endocrine responses, we hypothesized that fructose modifies the secretion of miR-143-5p, miR-140-5p, miR-146b-5p, miR-223-3p, miR-21-5p, miR-342-3p, miR-148a-5p and miR-450a-5p through EVs derived from 3T3-L1 cells and rat plasma. Thus, the aim of this study was to evaluate the effect of fructose exposure on the secretion of selected miRNAs in extracellular vesicles from 3T3-L1 cells and plasma from Wistar rats.

## MATERIALS & METHODS

### 3T3-L1 cell culture

3T3-L1 cells were acquired from ATCC and cultured in six-well plates ($8 \times 10^4$ cells per well) with Dulbecco's Modified Eagles Medium (DMEM, Gibco, Grand Island, NY, USA) supplemented with 0.1 mM L-glutamine, 1 mM sodium pyruvate, 0.1 mM nonessential amino acids, 1% gentamicin and 10% fetal bovine serum (FBS) until they reached confluence. For EVs collection experiments, FBS was ultracentrifuged at $118,000 \times g$ for 18 h (45 Ti rotor, Beckman Coulter) to remove the EVs (*Shelke et al., 2014*). After reaching confluence, cells were incubated with differentiation cocktail: 0.5 mM 3-isobutyl-1-methylxanthine, 0.25 μM dexamethasone and 0.8 μM bovine insulin in DMEM with 10% FBS. Two days later, the culture medium was replaced with DMEM supplemented 10% EV-depleted FBS with 550 μM fructose (Sigma-Aldrich; MO, USA). A previous study demonstrated that 550 μM of fructose promotes adipogenesis in 3T3-L1 cells (*Du & Heaney, 2012*). Afterward, the culture medium was replaced every two days until the fourth day when the culture medium was recovered, and cells were harvested. All cultures were maintained under humidified conditions and incubated at 5% $CO_2$ and 37 °C.

### Animals

Rats were raised in the animal facilities at Instituto Nacional de Cardiología Ignacio Chávez and were handled following the regulations of the Mexican Official Norm (NOM-062-ZOO-1999) for production, care, and use of laboratory animals. Twenty male Wistar rats of eight weeks of age and weighing 100–120 g were randomly allocated to two groups. All

**Table 1 Selection of microRNAs.**

| miRNA | Function | Reference |
|---|---|---|
| miR-21-5p | Increased in total plasma fructose-fed mice. Regulated adipogenesis in 3T3-L1 cells. Increased in adipose tissue of obese, in adipose tissue of T2DM patients. | *Kang et al. (2013)*, *Engin (2017)*, *Guglielmi et al. (2017)* |
| miR-146b-5p | Increased in total plasma of obese patients. Regulated adipogenesis in 3T3-L1 cells and regulate glucose metabolism in adipocytes. Increased in adipose tissue of obese. | *Chen et al. (2014)*, *Cui et al. (2018)* and *Zhu et al. (2018)* |
| miR-140-5p | Increase in total plasma of obese and T2DM. Regulated adipogenesis in 3T3-L1 cells. | *Ortega et al. (2013)*, *Ortega et al. (2014)*, *Wang et al. (2015a)* and *Li et al. (2017)* |
| miR-143-5p | Increased in total plasma of obese. Regulated adipogenesis in human adipocytes and 3T3-L1 cells, and regulated lipid accumulationin 3T3-L1 cells. | *Esau et al. (2004)*, *Xie, Lim & Lodish (2009)*, *Wang et al. (2011)* and *Quintanilha et al. (2019)* |
| miR-342-3p | Decreased in total plasma of obese and in total plasma of insulin-resistant subjects. Regulated adipogenesis in 3T3-L1 cells, and lipogenesis non-adipocyte cells. | *Li et al. (2013)*, *Wang et al. (2011)*, *Wang et al. (2015b)*, *Wang et al. (2015a)*, *Khalyfa et al. (2016)* and *Masotti et al. (2017)* |
| miR-223-3p | Decrease in total plasma of obese, in total plasma of T2DM patients, and increased in EVs of sucrose-fed rats. Regulated adipogenesis in mesenchymal stem cells. in insulin resistance human adipose tissue and inflammatory response. | *Zampetaki et al. (2010)*, *Bauernfeind et al. (2012)*, *Chuang et al. (2015)*, *Kilic et al. (2015)*, *Wen, Qiao & Wang (2015)*, *Guan et al. (2015)*, *Chen et al. (2014)* and *Brianza-Padilla et al. (2016)* |
| miR-450a-5p | Regulated adipogenesis in 3T3-L1 cells mediated by exosome-like vesicles. Increased in insulin resistance in HUVEC cells. | *Zhang et al. (2017)* and *Wei, Meng & Zhang (2020)* |
| miR-148a-5p | Regulated adipogenesis in 3T3-L1 cells. Decrease in inflammatory response in human adipose tissue-derived mesenchymal stem cells. | *Shi et al. (2015a)*, *Shi et al. (2015b)* and *Shi et al. (2016)* |

animals were housed under artificial 12-hour light/dark cycles and a temperature of 22 °C. The control group allocated in a cage ($n = 6$) had free access to tap water, and the fructose group ($n = 14$) allocated in three cages (five rats per cage) had to access to a 20% fructose solution (w/v) as their only liquid source. Both groups received a standard rodent diet (Laboratory Rodent Diet 5001: protein 24.1%, fat 11.4%, fiber (crude) 5.2% carbohydrates 48.7%; Starch 21.9% sucrose 3.15%, for Nutrition International, Brentwood, MO, USA) for eight weeks. Rats were given ad libitum access to diet. The euthanasia was carried out by an injection of pentobarbital is approved by NOM-062-ZOO-1991. The main project was approved by the Internal Animal Care and Use Committee of Instituto Nacional de Cardiología Ignacio Chávez (Permit No INC/CICUAL/009/2018).

## Biochemical measurements

After eighth weeks, all rats were weighed, fasted for six hours, and sacrificed. Blood samples were collected by cardiac puncture under terminal anesthesia (pentobarbital), using K+EDTA as an anticoagulant. Epididymal adipose tissue was collected and weighed. Plasma

was obtained by blood centrifugation (2000 × g for 15 min at 4 °C) and stored at −70 °C until needed. Glucose (DCL- glucose oxidase Diagnostic Chemical Limited de Mexico, Mexico), triglycerides, total cholesterol (SPINREACT cholesterol-LQ and triglycerides-LQ; Spinreact S.A. Girona, Spain) and HDL-C (Hitachi 902 analyzer: Hitachi LTD, Tokyo, Japan) were determined using standard enzymatic procedures. The accuracy and precision of the biochemical measures are provided in a previous report (*Brianza-Padilla et al., 2016*). Insulin resistance was estimated using the homeostasis model assessment method (HOMA). It was calculated using the following formula: Plasma glucose (mg/dL) × fasting plasma insulin (IU mg/L) in the fasting state divided by 405 (*Matthews et al., 1985*).

## Cytokines and Insulin Resistance Measurements

Plasma leptin, insulin, and IL-1β were determined using Milliplex MAP rat adipokine magnetic bead panel kit (Millipore; Billerica, MA, USA) following the manufacturer's indications. Proteins were analyzed using a Luminex MAGPIX system (Luminex Corporation; Houston, TX, USA) and Milliplex Analyst software (Millipore; St. Charles, MO, USA). Plasma adiponectin was determined using the ELISA kit for Rat Adiponectin (Millipore; Billerica, MA, USA) following the manufacturer's indications.

## Measurement of adipocyte size

Epididymal adipose tissue fixed in 10% formalin (Sigma-Aldrich; MO, USA) was used for hematoxylin and eosin staining. The adipocyte area of epididymal adipose tissue was analyzed using ImageJ software (National Institutes of Health, Bethesda, MD, USA) from hematoxylin and eosin-stained adipose tissue sections. The mean adipocyte surface area was calculated from 500 cells/rat.

## RNA Isolation of 3T3-L1 preadipocytes and rat epididymal adipose tissue

Total RNA was extracted from 3T3-L1 cells and 100 mg of rat epididymal adipose tissue by Tripure method following the manufacturer's protocol (Roche; Basel, Switzerland). miR-143-5p, miR-140-5p, miR-146b-5p, miR-223-3p, miR-21-5p, miR-342-3p, miR-148a-5p and miR-450a-5p were determined using two-step RT-qPCR with RT-primer specific assay in combination with TaqMan probes (Applied Biosystems; CA, USA). The RNA isolated was immediately converted to cDNA, as described below.

## Extracellular vesicles size and number estimation

NanoSight NS300 was used to determine vesicle size and concentration (Malvern Instruments Ltd; Malvern, UK). Previously, the supernatant or plasma was centrifuged (10,000 × g for 30 min at 4 °C) to remove cell debris, and the supernatant was recovered. Briefly, the samples were diluted 4.5:500 in PBS, and each sample was injected into the NanoSight chamber. The camera gain was set at a constant value of 10, and the threshold value for vesicle detection was set at 5.

## Western blot analysis

After the fourth day of fructose treatment, cell culture media was collected and centrifuged at 400 × g for 10 min to remove cells. The supernatant was centrifuged at 2,000 × g for

20 min to remove cell debris, and the resultant supernatant was stored at $-70\,°C$ until further use. The samples were thawed at $4\,°C$. Two mL of a PEG 8,000 solution (50% w/v) was added per each 10 mL of sample and incubated overnight at $4\,°C$. The samples were centrifuged at $1,500 \times g$, and the pellet was resuspended in 2.7 mL of PBS. The sample was pipetted into $13 \times 51$ mm tubes for ultracentrifugation (Optima MAX, Beckman Coulter) at $118,000 \times g$ (53,000 rpm, k-factor 56.7) at $4\,°C$ in a fix angle rotor (TLA 100.3, Beckman Coulter; CA, USA) for 39 min. The resultant pellet was resuspended in 50 µL of RIPA buffer supplemented with 1x protease inhibitors cocktail and 1x EDTA (Halt Protease Inhibitor Single-Use Cocktail, Thermo Scientific; MA, USA). A Tricine-SDS-PAGE method was used to separate proteins. After blotting, membranes were incubated overnight with rabbit anti-ANXA2 (Abcam, ab178677) at 1:5,000 and rabbit anti-CD63 (Abcam, ab193349) at 1: 2,000.

## RNA Isolation of extracellular vesicles of supernatants of preadipocytes and rat plasma

600 µL of cell supernatants and rat plasma were centrifuged ($10,000 \times g$ for 30 min at $4\,°C$) to remove cell debris, and the supernatant was recovered. For RNA isolation of EVs the exoRNeasy serum/plasma midi kit (Qiagen; Hilden, Germany) was used. During the RNA purification step, the same amount of cel-miR-39 spike-in control was added (Qiagen; Hilden, Germany) according to the provider recommendations and previous publication (*Enderle et al., 2015*). The RNA isolated from EVs was immediately converted to cDNA, as described below.

## miRNAs determination by RT-qPCR

miRNAs were determined using two-step RT-qPCR with RT-primer specific assay in combination with TaqMan probes: miR-143-5p (Assay ID: 463509_mat), miR-140-5p (Assay ID: 001187) miR-148a-5p (Assay ID: 473012_mat), miR-450a-5p (Assay ID: 462729_mat), miR-21-5p (Assay ID: 000397), miR-146b-5p (Assay ID: 002755), miR-342-3p (Assay ID: 002260), and miR-223-3p (Assay ID: 07896_mat) (Applied Biosystems; CA, USA). Each RT-reaction used 1.5 µL from the 14 µL eluted RNA using the TaqMan MicroRNA Reverse Transcription Kit (Applied Biosystems; CA, USA). The RT reaction program and PCR cycling conditions were as we previously reported (*Brianza-Padilla et al., 2016*). miRNAs relative concentrations were normalized with Ct values of cel-miR-39, and values were calculated using $2^{-\Delta\Delta Ct}$ and $2^{-\Delta Ct}$ formulas. All Ct values for cel-miR-39 ranged from 20 to 22 cycles both for total plasma and for EVs RNA isolations.

## Statistical analysis

Data are presented as means and standard errors. Data were tested for normality and equal variances. Accordingly, differences between groups were assessed by unpaired $t$-test or Mann–Whitney U test ($p < 0.05$) using the Graph Pad Prism software version 8.

## RESULTS

### Fructose exposure modified the release of microparticles in 3T3-L1 supernatant

To determine whether fructose exposure modified the release of EVs, we quantified the total number of vesicles and their size by nanoparticle tracking analysis. Fructose exposure increased the total number of particles in the cell culture media by two-fold as compared to the control group ($p = 0.0001$) (Fig. 1A). Although the mean size of EVs did not change between groups, we observed that fructose favored the release of vesicles below 200 nm (Fig. 1B). Finally, the proteins characteristic of EVs such as CD63 and CD81 (tetraspanins enriched in late multivesicular bodies) (*Kowal, Tkach & Théry, 2014*) and ANXA2 (calcium-dependent phospholipid-binding protein, as one of the most highly expressed proteins in EVs) (*Valapala & Vishwanatha, 2011*) were determined by western blot. As expected, we found the presence of CD63, ANXA2, and CD81 in EVs of the 3T3-L1 cell supernatant (Fig. 1C).

### Fructose modifies miRNA levels in supernatant EVs and 3T3-L1 cells

In this study, we used a high concentration of fructose according to a previous study (*Du & Heaney, 2012*) to evaluate the effects of fructose on miRNAs expression on EVs and cells, we exposed 3T3-L1 cells at 550 µM of fructose for four days. We showed that exposure to 550 µM of fructose promoted high lipid accumulation compared to basal conditions ($p = 0.0022$) (Fig. S1A). Moreover, *Ppparg* ($p = 0.05$), *Glut4* ($p = 0.05$) and *Cebpa* ($p = 0.05$) expression were upregulated in both culture conditions compared to basal group (Fig. S1B). Protein levels of PPARγ and GLUT4 increased in both control ($p = 0.0140$) and fructose exposure compared with basal group (Fig. S1C). On the other hand, the differences in miRNA abundance is shown in Table 2, being miR-21-5p > miR-143-5p > miR-140-5p > miR-146b-5p > miR-342-3p > miR-223-3p > miR-148a-5p > miR-450a-5p in EVs from supernatans. EVs from fructose stimulated cells showed 3.8 times more miR-143-5p levels than the EVs derived from the 3T3-L1 cells of the control group ($p = 0.0286$) (Table 2). In contrast, we found a decrease of miR-223-3p levels (6-fold change) in EVs compared to the control cells ($p = 0.0286$) (Table 2). No differences were observed in miR-342-3p, miR-140-5p, miR-21-5p, miR-148a-5p, miR-450a-5p and miR-146b levels in EVs (Table 2). The differences in miRNA abundance in cell extracts are shown in Table 2, being miR-21-5p > miR-146b-5p > miR-140-5p > miR-143-5p > miR-342-3p > miR-223-3p > miR-450a-5p > miR-148a-5p. Fructose increased the expression of miR-223-3p ($p = 0.0079$), miR-143-5p ($p = 0.0179$), miR-140-5p ($p = 0.0079$), miR-146b-5p ($p = 0.0411$) and miR-342-3p ($p = 0.0159$) at a cellular level (Table 2), while the expressions of miR-21-5p, miR-148a-5p and miR-450a-5p did not change (Table 2).

### Fructose intake in water regulates metabolic parameters, miRNA levels in EVs plasma and rat adipose tissue

In this study, we used a physiological concentration of fructose according to previous studies (*Goran, Ulijaszek & Ventura, 2013*). We found that fructose exposure did not

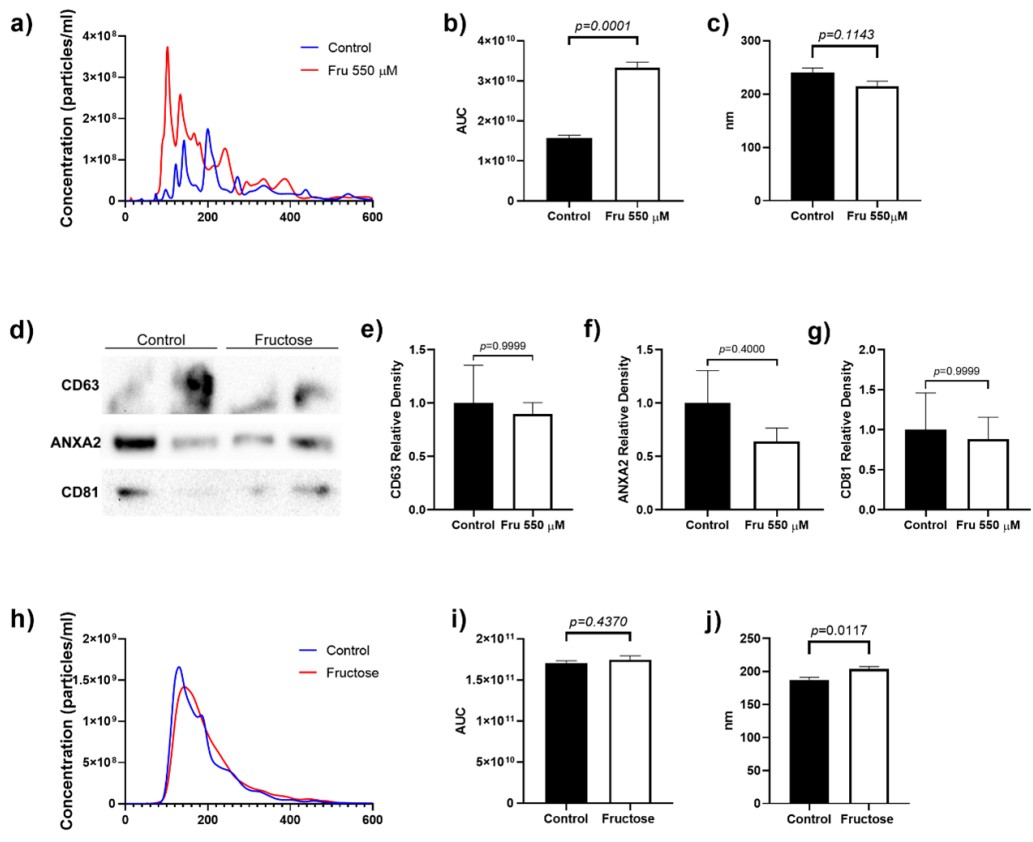

**Figure 1** **Particle number estimation and size of EVs derived from supernatants from 3T3-L1 cells and rat plasma.** (A) The total number of particles from 3T3-L1 cells supernatants was estimated by nanoparticle tracking analysis. (B) Area under the curve from the total number of particles from 3T3-L1 cells supernatants. (C) The mean size of EVs from 3T3-L1 cells supernatants. (D) Western blot for CD63, ANXA2, and CD81 in EVs of 3T3-L1 cell supernatants. (E) Relative density of protein of CD63. (F) Relative density of protein of ANXA2. (G) Relative density of protein of CD81. (H) The total number of particles from rat plasma was estimated by nanoparticle tracking analysis. (I) Area under the curve from the total number of particles from rat plasma. (J) The mean size of EVs from rat plasma. Differences were tested by the Mann–Whitney U test. Data are presented as means ± SE.

induce changes in total body weight and adipose tissue weight compared to the control group (Table 2) but increased the average of adipocyte area ($p = 0.039$) (Table 3). The fructose in the drinking water increased the levels of glucose ($p = 0.001$), triglycerides ($p = 0.001$), and insulin ($p = 0.007$) in plasma (Table 3). Similarly, the HOMA-IR index was higher in the fructose group than the control group ($p = 0.007$) (Table 3), as well as plasma adiponectin ($p = 0.017$), leptin ($p = 0.012$), and IL-1β ($p = 0.033$) levels (Table 3). In contrast, HDL-C levels were lower in the Fructose group than in the Control group ($p = 0.042$) (Table 3).

To evaluate whether chronic fructose exposure would modify miRNA expression in adipose tissue and plasma-derived EVs, we fed rats with 20% of fructose in drinking water for eight weeks. To determine the effect of fructose exposure in the amount of plasma-derived EVs and its molecular cargo, we first determined the microparticle concentration

**Table 2  Expression of adipocyte-related miRNAs in EVs and 3T3-L1 cells.**

| | Control ($n = 5$) | Ct Control | Fru 550 µM ($n = 5$) | Ct Fru 550 µM | Fold change | *P*-value | Ct *P*-value |
|---|---|---|---|---|---|---|---|
| **Extracellular vesicles** | | | | | | | |
| miR-21-5p | $0.0063 \pm 0.0014$ | $26.89 \pm 1.05$ | $0.0175 \pm 0.0079$ | $25.75 \pm 0.96$ | 2.777 | 0.2857 | 0.5556 |
| miR-146b-5p | $0.0002 \pm 4.21e^{-5}$ | $30.17 \pm 0.20$ | $0.0012 \pm 0.0009$ | $31.06 \pm 3.09$ | 6.000 | 0.6286 | 0.6286 |
| mir-140-5p | $0.0002 \pm 7.88e^{-5}$ | $30.38 \pm 0.45$ | $0.0014 \pm 0.0013$ | $29.64 \pm 1.42$ | 7.000 | 0.8571 | 0.8571 |
| mir-143-5p | $0.0009 \pm 0.0002$ | $29.76 \pm 1.09$ | $0.0034 \pm 0.0012$ | $27.92 \pm 1.06$ | 3.777 | 0.0286 | 0.3429 |
| miR-342-3p | $7.57e^{-5} \pm 1.37e^{-5}$ | $33.43 \pm 0.26$ | $0.0001 \pm 3.04e^{-5}$ | $32.88 \pm 0.29$ | 1.321 | 0.6286 | 0.2286 |
| miR-223-3p | $0.0006 \pm 0.0002$ | $31.15 \pm 0.31$ | $0.0001 \pm 4.14e^{-5}$ | $31.42 \pm 0.56$ | −6.000 | 0.0286 | 0.8857 |
| miR-450a-5p | $1.66e^{-6} \pm 1.01e^{-6}$ | $37.72 \pm 0.83$ | $5.75e^{-6} \pm 4.25e^{-6}$ | $36.42 \pm 0.89$ | 3.463 | 0.4000 | 0.4000 |
| miR-148a-5p | $1.33e^{-5} \pm 8.08e^{-6}$ | $34.71 \pm 0.82$ | $1.72e^{-5} \pm 5.39e^{-6}$ | $34.00 \pm 0.52$ | 1.293 | 0.4000 | 0.4000 |
| **3T3-L1 cells** | | | | | | | |
| miR-21-5p | $1.0000 \pm 0.0281$ | $18.18 \pm 0.06$ | $1.2290 \pm 0.2199$ | $18.24 \pm 0.02$ | 1.229 | 0.8413 | 0.5476 |
| miR-146b-5p | $0.1386 \pm 0.0016$ | $21.03 \pm 0.04$ | $0.1736 \pm 0.0198$ | $21.59 \pm 0.60$ | 1.252 | 0.0411 | 0.9999 |
| miR-140-5p | $0.0308 \pm 0.0010$ | $23.20 \pm 0.06$ | $0.0438 \pm 0.0072$ | $23.04 \pm 0.02$ | 1.422 | 0.0079 | 0.0556 |
| miR-143-5p | $0.0093 \pm 0.0004$ | $24.96 \pm 0.11$ | $0.0126 \pm 0.0011$ | $24.79 \pm 0.10$ | 1.354 | 0.0179 | 0.3929 |
| miR-342-3p | $0.0018 \pm 9.95e^{-5}$ | $27.25 \pm 0.10$ | $0.0031 \pm 0.0006$ | $26.93 \pm 0.04$ | 1.722 | 0.0159 | 0.0317 |
| miR-223-3p | $0.0007 \pm 6.16e^{-5}$ | $28.51 \pm 0.15$ | $0.0016 \pm 0.0005$ | $27.92 \pm 0.18$ | 2.285 | 0.0079 | 0.0079 |
| miR-450a-5p | $0.0003 \pm 1.71e^{-5}$ | $29.88 \pm 0.10$ | $0.0003 \pm 6.69e^{-5}$ | $29.87 \pm 0.09$ | 1.000 | 0.8889 | 0.9999 |
| miR-148a-5p | $4.60e^{-6} \pm 4.78e^{-7}$ | $35.94 \pm 0.15$ | $5.59e^{-6} \pm 8.58e^{-7}$ | $35.76 \pm 0.11$ | 1.215 | 0.7302 | 0.5556 |

Notes.

miRNAs expression was determined by RT-qPCR using cel-miR-39 as reference for EVs and U6 as reference for cells for the $2^{-\Delta Ct}$ method. Differences were tested by the Mann-Whitney U test. Data are presented as means $\pm$ SE.

**Table 3  Biochemical data in Control and Fructose group.**

| | Control ($n = 6$) | Fructose ($n = 14$) | *P*-value |
|---|---|---|---|
| Body weight (g) | $350.20 \pm 15.21$ | $341.9 \pm 5.92$ | 0.792 |
| Epididymal Adipose tissue weight (g) | $2.51 \pm 0.19$ | $2.95 \pm 0.14$ | 0.148 |
| Average of adipocyte area ($\mu m^2$) | $1620 \pm 52.29$ | $2090 \pm 105.6$ | 0.039 |
| Glucose (mg/dL) | $160.70 \pm 8.95$ | $232.50 \pm 17.89$ | 0.001 |
| Insulin (pg/mL) | $16.49 \pm 2.42$ | $44.9 \pm 4.65$ | 0.007 |
| HOMA-IR | $0.270 \pm 0.04$ | $0.870 \pm 0.15$ | 0.007 |
| Total Cholesterol (mg/dL) | $47.98 \pm 1.67$ | $48.92 \pm 1.29$ | 0.920 |
| Triglycerides (mg/dL) | $53.83 \pm 7.56$ | $125.30 \pm 16.15$ | 0.001 |
| HDL-Cholesterol (mg/dL) | $38.98 \pm 1.87$ | $33.88 \pm 1.51$ | 0.042 |
| Leptin (pg/mL) | $902.70 \pm 208.5$ | $1751 \pm 160.2$ | 0.012 |
| Adiponectin (ng/mL) | $52.42 \pm 3.08$ | $72.17 \pm 4.12$ | 0.017 |
| IL-1β (pg/mL) | $19.59 \pm 10.27$ | $107.70 \pm 29.59$ | 0.033 |

Notes.

Differences were tested by the Mann–Whitney U test. Data are presented as means $\pm$ SE.

by NTA analysis. The number of particles were similar between both groups of animals (Fig. 1D), while the mean size of EVs increased in fructose-fed rats compared to the Control group ($p = 0.0117$) (Fig. 1E). Despite the similar levels of plasmatic EVs, we observed that high fructose intake induced changes in miRNA content. The differences

in miRNA abundance are shown in Table 4, being miR-223-3p > miR-21-5p > miR-140-5p > miR-143-5p > miR-146b-5p > miR-342-3p > miR-450a-5p > miR-148a-5p in EVs from plasma. The miR-143-5p levels increased 10.14-fold in EVs of the fructose rats ($p$ = 0.0010) (Table 4), while miR-223-3p (−3.8 fold change) ($p$ = 0.0220), miR-342-3p (−9.47 fold change) ($p$ = 0.0011), miR-140-5p (−4.65 fold change) ($p$ = 0.0129) and miR-146b-5p (−4.27 fold change) ($p$ = 0.0245) levels were reduced (Table 4). We did not observe changes in miR-21-5p, miR148a-5p and miR-450a-5p levels in EVs (Table 4). Interestingly, we found a positive correlation between levels of miR-143-5p in EVs and triglycerides (rho = 0.7098; $p$ = 0.005) and a negative correlation between levels of miR-143-5p in EVs and HDL-C (rho = −0.4577; $p$ = 0.0425) (Table S1). The levels of miR-342-3p in EVs negatively correlated with triglycerides (rho = −0.3895; $p$ = 0.0448) (Table S1). The levels of miR-148a-5p in EVs positively correlated with glucose (rho = 0.6657; $p$ = 0.0014) (Table S1). Finally, the levels of miR-146b-5p in EVs negatively correlated with leptin (rho = −0.4617; $p$ = 0.0405) (Table S1). The differences in miRNA abundance in adipose tissue is shown in Table 4, being miR-21-5p > miR-223-3p > miR-450a-5p > miR-140-5p > miR-146b-5p > miR-143-5p > miR-342-3p > miR-148a-5p. In the adipose tissue, we found that fructose exposure induced an increase in miR-143-5p expression ($p$ = 0.0143) (Table 4), whereas miR-223-3p ($p$ = 0.0462) and miR-342-3p ($p$ = 0.0320) expression were reduced in the Fructose group (Table 4). We did not find differences in the expression of miR-140-5p, miR-21-5p, miR-148a-5p, miR-450a-5p and miR-146b-5p (Table 4).

## Pathway enrichment of adipocyte miRNAs

To predict the cellular pathways targeted during fructose exposure, we used DIANA-miRPath for identification of putative miRNA targets. The analysis Kyoto Encyclopedia of Genes and Genomes (KEGG) pathway from miRPath revealed that miRNAs altered by fructose exposure target genes involved in TGF-$\beta$ and mTOR signaling pathways, each signaling pathway have putative targets of 25 genes (Figs. S2, S3 and Table S4).

## DISCUSSION

In our study, we found that high fructose exposure modified the abundance of some adipocyte-related miRNAs in EVs derived from 3T3-L1cells supernatants and rat plasma. In particular, fructose exposure increased the levels of miR-143-5p and decreased miR-223-3p levels in EVs. In rat plasma, fructose promoted the production of large EVs size and increased the levels of miR-143-5p in EVs, and reduced miR-223-3p, miR-140-5p, and miR-342-3p.

In this report, we showed that 20% (w/v) of fructose in drinking water for eight weeks did not change body weight but increased in the average of adipocyte area without changing in the total adipose tissue weight. These results, together with the increase in glucose, insulin, and triglyceride levels and a decrease in HDL-C levels, suggest the development of the metabolic syndrome. Similar findings have been found in a previous report (*Zubiría et al., 2016*).

**Table 4  Expression of adipocyte-related miRNAs in EVs and adipose tissue.**

| | Control ($n = 6$) | Ct Control | Fructose ($n = 14$) | Ct Fructose | Fold change | *P*-value | Ct *P*-value |
|---|---|---|---|---|---|---|---|
| **Extracellular vesicles** | | | | | | | |
| miR-21-5p | $0.8459 \pm 0.1206$ | $21.55 \pm 0.28$ | $0.3688 \pm 0.1570$ | $24.18 \pm 0.60$ | $-2.293$ | 0.1093 | 0.1528 |
| miR-223-3p | $4.9440 \pm 0.4456$ | $18.95 \pm 0.14$ | $1.2750 \pm 0.6552$ | $23.01 \pm 0.70$ | $-3.877$ | 0.0220 | 0.0167 |
| miR-450a-5p | $0.0006 \pm 0.0001$ | $32.13 \pm 0.49$ | $0.0005 \pm 9.632e^{-5}$ | $30.89 \pm 0.14$ | $-1.200$ | 0.7181 | 0.0064 |
| miR-140-5p | $0.0810 \pm 0.0099$ | $24.89 \pm 0.15$ | $0.0174 \pm 0.0100$ | $27.37 \pm 0.38$ | $-4.655$ | 0.0129 | 0.0135 |
| miR-146b-5p | $0.0235 \pm 0.0021$ | $26.66 \pm 0.14$ | $0.0055 \pm 0.0036$ | $30.88 \pm 0.68$ | $-4.272$ | 0.0245 | 0.0245 |
| miR-143-5p | $0.0014 \pm 0.0005$ | $31.42 \pm 0.86$ | $0.0142 \pm 0.0017$ | $26.22 \pm 0.51$ | 10.142 | 0.0010 | 0.0010 |
| miR-342-3p | $0.0483 \pm 0.0046$ | $25.62 \pm 0.13$ | $0.0051 \pm 0.0041$ | $28.92 \pm 0.35$ | $-9.470$ | 0.0011 | 0.0011 |
| miR-148a-5p | $1.714e^{-5} \pm 4.490e^{-6}$ | $37.42 \pm 0.57$ | $3.024e^{-5} \pm 5.187e^{-6}$ | $35.17 \pm 0.29$ | 1.764 | 0.1250 | 0.0010 |
| **Adipose tissue** | | | | | | | |
| miR-21-5p | $0.4912 \pm 0.0749$ | $20.17 \pm 0.21$ | $0.4342 \pm 0.0378$ | $20.04 \pm 0.11$ | $-1.131$ | 0.5214 | 0.8314 |
| miR-223-3p | $0.1674 \pm 0.0200$ | $21.69 \pm 0.13$ | $0.1297 \pm 0.0061$ | $21.74 \pm 0.06$ | $-1.290$ | 0.0462 | 0.8983 |
| miR-450a-5p | $0.0299 \pm 0.0024$ | $24.15 \pm 0.07$ | $0.0297 \pm 0.0018$ | $23.88 \pm 0.07$ | $-1.004$ | 0.8314 | 0.0462 |
| miR-140-5p | $0.0301 \pm 0.0058$ | $24.24 \pm 0.23$ | $0.0278 \pm 0.0019$ | $23.99 \pm 0.10$ | 1.083 | 0.8983 | 0.4155 |
| miR-146b-5p | $0.0118 \pm 0.0019$ | $25.55 \pm 0.22$ | $0.0112 \pm 0.0006$ | $25.28 \pm 0.07$ | $-1.050$ | 0.7166 | 0.2818 |
| miR-143-5p | $0.0118 \pm 0.0008$ | $25.42 \pm 0.16$ | $0.0214 \pm 0.0030$ | $24.45 \pm 0.17$ | 1.816 | 0.0143 | 0.0034 |
| miR-342-3p | $0.0086 \pm 0.0009$ | $25.96 \pm 0.21$ | $0.0066 \pm 0.0011$ | $26.14 \pm 0.22$ | $-1.305$ | 0.0320 | 0.2908 |
| miR-148a-5p | $0.0001 \pm 3.925e^{-5}$ | $31.81 \pm 0.30$ | $0.0001 \pm 1.369e^{-5}$ | $31.76 \pm 0.13$ | $-1.230$ | 0.6865 | 0.8314 |

**Notes.**

miRNAs expression was determined by RT-qPCR using cel-miR-39 as reference for EVs and U6 as reference for cells for the $2^{-\Delta Ct}$ method. Differences were tested by the Mann-Whitney U test. Data are presented as means $\pm$ SE.

The dose of fructose was chosen to resemble the intake of the highest consumers of added sugars in the USA which represents 20–25% of daily caloric intake (*Goran, Ulijaszek & Ventura, 2013*). Moreover, the concentration of fructose in beverages often vary between 10–30% (w/v), and the duration of these experiments is also variable (*Toop & Gentili, 2016*). Therefore, there is not a consensus about a particular experimental concentration of fructose intake. Moreover, studies in healthy humans showed that the intake of a fructose 20% (w/v) solution increased plasma triacylglycerol and glucose concentrations (*Faeh et al., 2005*; *Lê et al., 2006*). Based in this information, we chose a concentration of fructose 20% in drinking water to assure the induction of metabolic disorders. The increase of the average of adipocyte area suggest that the exposure of fructose for eight weeks induces hypertrophy of the adipocytes. This effect may be explained because the fructose fed rats developed hyperleptinemia. It has been reported that the hyperleptinemia and leptin resistance in adipose tissue induced by chronic fructose intake favors adipocyte hypertrophy (*Sangüesa et al., 2018*). Furthermore, hypertrophic adipocytes may release paracrine factors, including EVs, which promote the recruitment of preadipocytes and induce their differentiation into mature adipocytes.

Current studies had demonstrated that EVs are endocrine signals that regulate several cellular processes. Several studies found that adipocytes can release EVs under different stimuli (*Connolly et al., 2015*; *Durcin et al., 2017*). For example, it has been observed that 3T3-L1 cells release EVs through the stimulation of cAMP/Epac-dependent, and this release is augmented by a combination of Ca2+ and ATP (*Komai et al., 2014*). Moreover,

the long-chain omega-3 fatty acid docosahexaenoic acid promotes the increase in the release of EVs in 3T3-L1 cells (*Declercq, D'Eon & McLeod, 2015*). Our findings showed that fructose exposure increased the release of EVs from supernatants of 3T3-L1 cells, but the amount of EVs did not change in the plasma of fructose-fed rats. We think that the increased amount of released EVs from cell cultures could result from the direct to exposure to high fructose concentration (550 µM). In contrast, some studies have reported that after the ingestion of sweetened drinks, the systemic concentration of fructose is of 363.4 µM (*Le et al., 2012*). Similarly, rats that had access to a solution rich in fructose (2g/kg) displayed a concentration of 146 µM of fructose in peripheral blood (*Sugimoto et al., 2010*). Therefore, it is possible that adipose tissue in vivo is exposed to lower concentrations of fructose. Also, the effect of fructose could be obscured in plasma-derived EVs because they correspond a complex mixture from different cell types. In the future it would be necessary to specifically isolate EVs from adipose tissue.

In a recent study, it was shown that adipocytes release two types of EVs: small EVs (below 100 nm) and large EVs (100–200 nm) (*Durcin et al., 2017*). In this study, we found that fructose exposure promoted the release of EVs between 100–200 nm. Thus, our results suggest that fructose induces the release of large EVs in 3T3-L1 cells, which may be enriched in adipogenic signaling proteins and mitochondrial markers (*Durcin et al., 2017*). Further studies are needed to elucidate whether fructose promotes the release of EVs of adipose tissue origin in rat plasma that has a metabolic role. On the other hand, currents findings have demonstrated that the RNA-binding proteins (RBPs) are involved in sorting the miRNAs process into EVs (*Groot & Lee, 2020*). Among these proteins, it was found the Y-box Binding Protein 1 promotes the sorting of miR-223-3p into EVs from HEK293T cells (*Shurtleff et al., 2016*). Because the fructose exposure modified levels of miRNAs in EVs from adipocytes, likely fructose regulates RBPs involved in the sorting process into EVs. However, more studies are needed to elucidate the effect of fructose on the modulation of miRNAs levels in EVs by regulation of RBPs.

In our results, we found that fructose increased miR-143-5p expression from EVs of both 3T3-L1 cells and rat plasma as well as from whole tissue and cells extracts (Fig. 2).The miR-143 is conserved between rats and humans, and several studies show an important role for this miRNA in lipid metabolism, adipogenesis, and insulin resistance. Similarly, augmented levels of miR-143-5p expression were reported in vitro models of adipogenesis (*Esau et al., 2004*). Also, this miRNA induces adipogenesis by promoting triglycerides synthesis (*Wang et al., 2011*) and accelerates lipid accumulation (*Xie, Lim & Lodish, 2009*). We showed that fructose induced an elevation of the levels of miR-143-5p in EVs derived from both 3T3-L1 cells and rat plasma, which supports an active metabolic role for EVs in metabolic processes. In agreement with this notion, we found a positive correlation between levels of miR-143-5p in EVs of fructose-fed rats and triglycerides and a negative correlation with HDL-C. Moreover, obese subjects and healthy subjects with a high-fat diet also presented high levels of miR-143-5p in the total plasma (*Quintanilha et al., 2019*; *Ramzan et al., 2019*). Recent studies showed that miR-143-5p remains unchanged in the total plasma of sucrose-fed mice for 20 weeks (*Hanousková et al., 2019*) and sucrose-fed rats for 4 weeks (*Yerlikaya & Mehmet, 2019*). Conversely, we also found that fructose-fed

Peer J

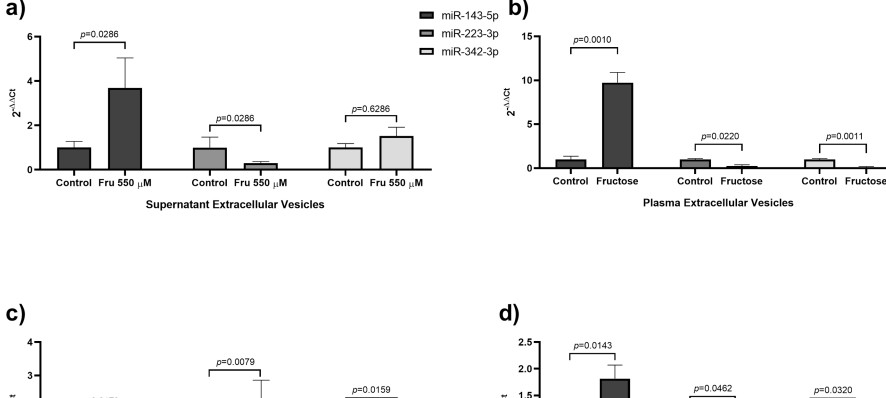

**Figure 2** **Fructose exposure modified the levels of miR-143-5p and miR-223-3p in EVs, 3T3-L1 cells, and adipose tissue.** (A) Levels of miR-143-5p (black), miR-223-3p (grey) and miR-342-3p (light grey) in EVs from 3T3-L1 cells supernatant. (B) Levels of miR-143-5p (black), miR-223-3p (grey) and miR-342-3p (light grey) in EVs from rat plasma. (C) Expression of miR-143-5p (black), miR-223-3p (grey) and miR-342-3p (light grey) in 3T3-L1 cells. (D) Expression of miR-143-5p (black), miR-223-3p (grey) and miR-342-3p (light grey) in rat adipose tissue. Differences were tested by the Mann–Whitney U test. Data are presented as means $\pm$ SE.

rats for 4 weeks increase the levels of miR-143-5p in total plasma ($p = 0.031$) (Table S3) together with an elevation of triglycerides ($p = 0.005$) (Table S2). The fructose exposure increased ten-fold the abundance of miR-143-5p in EVs. Our data supports the notion that fructose may induce lipid metabolism by endocrine mechanisms, including the release of EVs containing miR-143-5p.

Our study showed that the exposure to fructose decreases in miR-223-3p levels in EVs from both cell supernatants and rat plasma (Fig. 2). Opposite results were found in EVs of sucrose-fed rats for a longer period of time (six months) (*Brianza-Padilla et al., 2016*). This difference could be due to the source of fructose (sucrose). Previous reports showed that sucrose and the mix of monosaccharides (fructose+glucose) have different effects on lipid and glucose metabolism in rats and humans. Therefore, the low miR-223-3p levels in EVs of rat plasma could be an effect of the exposition to only fructose (*Sánchez-Lozada et al., 2010*; *Sheludiakova, Rooney & Boakes, 2012*; *Evans et al., 2017*). The miR-223 is a miRNA conserved between rats and humans. Moreover, miR-223-3p is reduced in total plasma of obese subjects (*Kilic et al., 2015*; *Wen, Qiao & Wang, 2015*) and T2DM patients (*Zampetaki et al., 2010*). The function of miR-223-3p has been related to the inflammatory response and adipocyte differentiation. Studies had demonstrated that miR-223-3p promotes adipogenesis in mesenchymal stem cells (*Guan et al., 2015*) and human adipocytes (*Liu et al., 2010*). Our study showed that the exposure to fructose augmented miR-223-3p expression in 3T3-L1 cells, while it was reduced in rat adipose tissue (Fig. 2). The differential miRNA expression between cell culture and tissue could be

due to the cell complexity of adipose tissue (*Kanneganti & Dixit, 2012*). For example, in visceral adipose tissue from obese patients, the miR-223-3p was upregulated specifically in the stromal vascular cells, while in adipocytes, this miRNA was unchanged (*Deiuliis et al., 2016*). Additionally, the upregulation of miR-223-3p leads to the downregulation of GLUT4 in human adipocytes (*Chuang et al., 2015*). On the other hand, in mice adipose tissue, miR-223-3p suppresses pro-inflammatory activation of macrophages (*Zhuang et al., 2012*) and regulates the production of NLRP3 and IL-1β (*Bauernfeind et al., 2012*). Although a role for miR-223-3p in adipogenesis is suggested by its increase in 3T3-L1 cells treated with fructose. In whole adipose tissue, fructose may induce an IL-1β-mediated inflammatory response due to the downregulation of miR-223-3p. However, additional studies are necessary to elucidate whether systemic and local downregulation of miR-223-3p in EVs participates in systemic IL-1β production.

Our study showed that fructose reduced miR-342-3p levels in EVs from rat plasma (Fig. 2). A similar effect has been observed in total plasma of obese children (*Khalyfa et al., 2016*) and insulin-resistant subjects (*Wang et al., 2015a*; *Masotti et al., 2017*). The miR-342-3p is a miRNA conserved between rats and humans and is implicated in lipid metabolism and adipogenesis. In human adipose-derived mesenchymal stem cells, miR-342-3p enhances adipogenesis (*Wang et al., 2015b*). Also, in non-adipocyte cells, it was found that this miRNA reduces lipogenesis by inhibiting the expression of SREBP (*Li et al., 2013*). Interestingly, we found a negative correlation between levels of miR-342-3p in EVs from fructose-fed rats and triglycerides. Despite the abundance of miR-342-3p in EVs is medium, the fructose exposure induced a decrease of nine-fold of this miRNA. Therefore, our findings add to the notion that fructose may induce lipogenesis by the reduction of this miRNA in EVs and tissue. Moreover, we found that fructose increased miR-342-3p expression in 3T3-L1 cells and reduced in rat adipose tissue (Fig. 2). Because our findings show that fructose exposure is related to an increase of miR-342-3p in 3T3-L1 cells and that chronic fructose intake in rats decreases miR-342-3p in adipose tissue, and EVs may be due to that this miRNA is more expressed in adipocytes than the stromal vascular fraction from adipose tissue (*Oger et al., 2014*), so that other cell types contribute to this miRNA secretion.

In our results, fructose reduced the levels of miR-140-5p in EVs of rat plasma. Elsewhere a similar trend was found in the total plasma of obese patients (*Wang et al., 2015a*). However, other studies in morbidly obese subjects and diabetic subjects showed high levels of miR-140-5p in total plasma (*Ortega et al., 2013*; *Ortega et al., 2014*). Moreover, we found that fructose increased miR-140-5p expression in 3T3-L1 cells, supporting the notion that it is related to adipogenesis (*Li et al., 2017*). The abundance of miR-140-5p is medium in EVs, adipocytes, and adipose tissue and fructose exposure induced a reduction of the abundance of 4-fold of this miRNA in EVs from rat plasma. Thus, our results suggest that the miR-140-5p may be involved with metabolic alterations developed in fructose-fed rats. However, future experiments should document longer exposures to fructose to determine if chronic stimulation modifies levels in EVs.

This study showed that fructose exposure reduces the levels of miR-146b-5p in EVs of rat plasma. The miR-146b is a miRNA conserved between rats and humans. A study

in obese children and adult T2DM showed high miR-146b-5p levels in the total plasma, which was found that participated in the suppression of high concentration glucose-induced pancreatic insulin secretion (*Cui et al., 2018*; *Mohany et al., 2020*). Interestingly, we found a negative correlation between the levels of miR-146b-5p in EVs and leptin in plasma. Noteworthy, the leptin mRNA is a target of miR-146b-5p in breast adipose tissue (*Al-Khalaf et al., 2017*). Our findings showed that the abundance of miR-146b-5p in EVs is medium, and the fructose exposure reduced 4-fold of this miRNA in EVs. If this reduction may participate in the hyperleptinemia found in fructose-fed rats should be further investigated. On the other hand, we found that fructose increases miR-146b-5p expression only in 3T3-L1 cells. The miR-146b-5p is a miRNAs related to inflammatory response and adipogenesis. In this line, a previous study showed that miR-146b-5p induces adipogenesis in human preadipocytes (*Chen et al., 2014*). The same report found that the expression of this miRNA increases in adipose tissue of diet-induced obesity mice (*Chen et al., 2014*). Moreover, TNF-$\alpha$ treatment induces miR-146b-5p expression in adipocytes. Also, it has been demonstrated that this miRNA regulates glucose homeostasis in porcine primary preadipocytes by targeting IRS1 (*Zhu et al., 2018*). Therefore, our results on 3T3-L1 cells could indicate that fructose may promote adipogenesis by increasing miR-146b-5p expression.

The miR-450a-5p, miR-148a-5p and miR-21-5p participate in adipocyte differentiation. For example, Zhang and collaborators showed that miR-450a-5p induces adipogenesis through the transfer of this miRNA by EVs in adipose tissue-derived stem cells (*Zhang et al., 2017*). However, we found that the abundance of miR-450a-5p in EVs is low, and we did not find changes in this miRNA in both cells and rats. However, we found that fructose-fed rats for 4 weeks increase the levels of miR-450a-5p in total plasma (Table S3). A recent study showed that the up-regulation of this miRNA improves insulin resistance in non-adipocyte cells (*Wei, Meng & Zhang, 2020*). On the other hand, we showed that the abundance of miR-21-5p in EVs is high; however, we did not find changes in this miRNA in both cells and rats. It has been demonstrated that the presence of miR-21-5p is involved in the proliferation and differentiation of adipocyte precursors (*Jeong Kim et al., 2009*; *Kim et al., 2012*). Besides, miR-148a-5p is highly expressed in human mesenchymal stem cell-derived adipose tissue during adipogenesis (*Shi et al., 2015b*). We found that the abundance of miR-148a-5p is low in EVs, and fructose exposure did not change the expression of this miRNA, both in cells and rats. However, we found a positive correlation between the levels of miR-148a in EVs and glucose. This finding suggests that miR-148a-5p is associated with alterations in glucose metabolism. In addition, our findings suggest that fructose may also use other mechanisms that do not involve changes in miR-450a-5p, miR-148a-5p, and miR-21-5p expression.

Several miRNAs have been previously associated to regulate metabolic processes in humans and murine models. Through miRPath software, we predicted signaling pathways that could be affected in a combinatorial manner by exposure to fructose. Interestingly, miRNAs that change by fructose exposure were predicted to target 19 genes in the TGF-$\beta$ signaling pathway (Fig. 1S) (Table S4). In EVs from rat plasma, the decrease levels of miR-140-5p, miR-146b-5p, miR-223-3p and miR-342-3p could favor the TGF-β signaling

pathway by fructose exposure (Fig. 1S) (Table S4). Recent studies have reported that the activation of TGF-β signaling inhibits adipocyte differentiation (*Luo et al., 2019*) and is involved in lipid accumulation in the liver (*Qin et al., 2018*). Therefore, fructose could favor lipid accumulation by downregulating these EVs-associated miRNAs. The mTOR signaling pathway could also be regulating the selected miRNAs (Fig. 2S). The miRPath analysis predicted to target 21 genes in the mTOR signaling pathway (Table S4). Studies indicate that the mTOR pathway is involved in several cellular processes, including proliferation and glucose and lipid metabolism (*Mao & Zhang, 2018*; *Sangüesa et al., 2019*). Moreover, a fructose-dependent increase in mTORC1 activity drives translation of pro-inflammatory cytokines (*Jones et al., 2021*). Therefore, fructose exposure could modulate these signal pathways involved in physiology adipose tissue by regulating selected miRNAs.

## CONCLUSIONS

Fructose exposure promotes an increase in the release of EVs with increased levels of miR-143-5p and decreased levels of miR-223-3p. Also, fructose induces an increase of miR-143-5p, miR-223-3p, miR-140-5p, miR-342-3p and miR-146b-5p in adipocytes. Similarly, in rats, the high fructose intake induces an increase of miR-143-5p and a decrease of miR-223-3p and miR-342-3p in both EVs and adipose tissue. Finally, the in vitro and in vivo models show that fructose may modify the way miRNAs are shipped into EVs. It would be important in the future to evaluate whether the miR-143-5p and miR-223-3p in EVs could be potential biomarkers of fructose exposure in humans.

## ACKNOWLEDGEMENTS

The authors are grateful to Hugo Villamil-Ramirez from the UGEPAS-Facultad de Química, Universidad Nacional Autónoma de México (UNAM)/Instituto Nacional de Medicina Genómica (INMEGEN), Mexico City 14610, Mexico, for providing laboratory technical advisory.

### Funding

The research received financial support from a scholarship from the National Council of Science and Technology (CONACYT 492169) was received by Adrian Hernandez-Diazcouder. The costs of cell cultures were covered by Universidad Autónoma Metropolitana-Iztapalapa. Finally, the costs of qPCR experiments and publishing in open access were covered by the Instituto Nacional de Cardiología Ignacio Chávez. Eduardo Martínez-Martínez received funding support from the Consejo Nacional de Ciencia y Tecnología (CB-258589). The funders had no role in study design, data collection and analysis, decision to publish, or preparation of the manuscript.

### Grant Disclosures

The following grant information was disclosed by the authors:

National Council of Science and Technology: CONACYT 492169.
Universidad Autónoma Metropolitana-Iztapalapa.
Instituto Nacional de Cardiología Ignacio Chávez.
Consejo Nacional de Ciencia y Tecnología: CB-258589.

## Competing Interests

The authors declare there are no competing interests.

## Author Contributions

- Adrián Hernández-Díazcouder and Eduardo Martínez-Martínez conceived and designed the experiments, performed the experiments, analyzed the data, prepared figures and/or tables, authored or reviewed drafts of the paper, and approved the final draft.
- Javier González-Ramírez, Abraham Giacoman-Martínez, Guillermo Cardoso-Saldaña, Ricardo Márquez-Velasco, José L. Sánchez-Gloria, Yaneli Juárez-Vicuña, Guillermo Gonzaga, Laura Gabriela Sanchez-Lozada and Julio César Almanza-Pérez performed the experiments, authored or reviewed drafts of the paper, and approved the final draft.
- Horacio Osorio-Alonso performed the experiments, analyzed the data, prepared figures and/or tables, and approved the final draft.
- Fausto Sánchez-Muñoz conceived and designed the experiments, analyzed the data, prepared figures and/or tables, authored or reviewed drafts of the paper, and approved the final draft.

## Animal Ethics

The following information was supplied relating to ethical approvals (i.e., approving body and any reference numbers):

The main project was approved by the Internal Animal Care and Use Committee of Instituto Nacional de Cardiología Ignacio Chávez (Permit No INC/CICUAL/009/2018).

## Data Availability

The raw measurements are available in the Supplemental File.

## Supplemental Information

Supplemental information for this article can be found online at http://dx.doi.org/10.7717/peerj.11305#supplemental-information.

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
