# Peer review of "High fructose exposure modifies the amount of adipocyte-secreted microRNAs into extracellular vesicles in supernatants and plasma"

_PeerJ, doi:10.7717/peerj.11305_

## Round 0.1 · original submission · Major Revisions

As you will see, both reviewers found this work to be of potential interest, and I agree with their assertion that the area is important and the study is timely and of potential interest.

However, Reviewer-1 has raised a number of issues which I feel must be addressed at a revision. Hence, I have marked this as a ‘Major Revision’ as these are all important and essential revisions.

Please address all points, but in particular note the following:

1. Can you provide markers and data to show that the slightly modified adipocyte differentiation protocol you used is effective? This could be one or all of Oil Red-O staining, immunoblotting of markers like PPARs or GLUT4 etc.

2. Using existing data sets, can you comment on the expression of plausible candidate miRNAs in 3T3-L1 and rt adipocytes (Major point 1 of reveower-2)

3. The next two points should be easy to address using bioinformatics - these are important points and need to be discussed/elaborated upon in the revision.

4. The disparity between the figures and tables is a worry – please can you take time to address these and make sure the data shown are consistent with the text.

5. In the manuscript at line 301-303, you mention hypertrophy of adipocytes because there is an increase in total adipocyte area. However, the area of individual adipocyte was not mentioned. It would be better if you could discuss the total area vs size of individual adipocyte (% of large adipocyte or average of adipocyte area) based on your results.

5. Finally, and importantly, discussion of the level of fructose employed needs to be at the start of the manuscript results section. Readers need to be able to evaluate this and consider it in the context of known levels in humans/rodents etc.

Reviewer 1 ·

Basic reporting

Minor points:

- Table 1 and Table 3 should be presented the same way, since it is quite hard to follow accordingly with the text. Format is acceptable, but both tables should be presented the same way. For instance, presenting first miRNAs data in tissue and then in plasma (as in Table 3). I really appreciate the Ct values appearing alongside with Delta Ct calculations.

- It would be appreciated that Figure 1c would come together with a quantification plot.

- Careful! Throughout the text, the are words that appear together. Please revise spelling.

In 3T3-L1 cell culture:
- “Cells cultures” should be replaced for “cell culture”
- “SFB” should be replaced for “FBS”

Experimental design

- I am very interested in the methodology used to differentiate 3T3-L1. Based on my experience, these cells should be differentiating at least 8 days to reach enough amount of lipid droplets within the adipocytes. Nevertheless, you maintain the cells for 6 days, and at day 2 the differentiation cocktail is replaced for fructose. Do you have any image of the cells at day 6, or RedOil staining that confirms the differentiation success?

- Another point about methodology refers to the miRNA isolation from EVs of supernatants and rat plasma. Did you perform preamplification after retrotranscription?

- The dose of fructose was chosen based on previous studies. A time-course using different doses would be interesting to check whether miRNAs fluctuate their expression. Is the dose used resembling physiology? And are in vitro and in vivo approaches comparable in terms of fructose amounts?

Validity of the findings

No comments.

Additional comments

This manuscript is an interesting study about the modulation of miRNA expression patterns upon fructose exposure. Considering the huge impact that miRNAs can trigger at tissue level, this is an attractive study relating epigenetics and fructose intake. The research involves both in vitro and in vivo studies, enhancing the strength of the data presented. However, to strengthen these findings, I suggest several comments to take into account.

Major points:

- Considering that miRNAs are able to target mRNA at tissue level, it would be interesting to analyse the expression of plausible targets of your candidate miRNAs in 3T3-L1 cells and rats’ adipose tissue. In case you could validate a specific mRNA as a target of your candidate miRNAs, functional test such as luciferase assays would be highly enlightening.

- Why have you focused in these specific 8 miRNAs? Considering previous literature, there are more miRNAs which have been previously reported to be implicated in adipose tissue physiology. Have you considered to perform arrays to obtain a wider view of miRNA expression profiles? Is this a future line of research?

- How would you explain different patterns in cells and rats? Have you checked the homology between your candidate miRNAs between humans and rats? Could in vivo data be translated into humans to use plasma miRNAs as biomarkers or tissue miRNAs as therapeutic tools?

- Please revise Figure 2. Data presented do not match with the text and the tables. For instance, in the discussion you point that miR-223-3p expression in 3T3-L1 cells was augmented upon fructose, agreeing with data presented in Table 1. Nevertheless, Figure 2 shows a global decrease both in vitro and in vivo. The same with miR-342-3p.

- In discussion section, you point at discrepancies between studies referring to miRNA expression. Have you elucidated which could be the reason attending to the differences? You claim that fructose is inducing a decrease in miR-223-3p levels in EVs in rat plasma, while previous studies showed the opposite. Considering that the feeding was longer, it would make sense to decipher the miRNA fluctuation according to fructose exposure.

·

Basic reporting

This is an interesting article and it is structured logically.
Background knowledge is introduced thoroughly.
Figures and results are shown clearly and raw data is supplied.

Experimental design

This research is highly relevant to the journal.
The research question is well defined and the rigorous investigation is performed to a high technical and ethical standard.
Sufficient details are given in the methods to replicate.

Validity of the findings

The results are validated and the conclusion is well stated accordingly.

Additional comments

Overall this is article is well presented.
The research question is novel and it is properly investigated.

In line 301-303, you mentioned the hypertrophy of adipocytes because there is an increase in total adipocyte area. However, the area of individual adipocyte was not mentioned. It would be better if you could discuss on the total area vs size of individual adipiocyte (% of large adipocyte or average of adipocyte area) based on your results.

---

## Round 0.2 · accepted · Accept

Both the reviewer and I are satisfied that you have carefully addressed all points.

Reviewer 1 ·

Basic reporting

After revising this second version of the manuscript, I agree with all the comments and consequently, the manuscript is accepted in the present form, with no more comments.

Experimental design

After revising this second version of the manuscript, I agree with all the comments and consequently, the manuscript is accepted in the present form, with no more comments.

Validity of the findings

After revising this second version of the manuscript, I agree with all the comments and consequently, the manuscript is accepted in the present form, with no more comments.

Additional comments

After revising this second version of the manuscript, I agree with all the comments and consequently, the manuscript is accepted in the present form, with no more comments.